# Deepening the Understanding of Carbon Active Sites for ORR Using Electrochemical and Spectrochemical Techniques

**DOI:** 10.3390/nano14171381

**Published:** 2024-08-24

**Authors:** Jhony Xavier Flores-Lasluisa, Diego Cazorla-Amorós, Emilia Morallón

**Affiliations:** 1Department Química Física e Instituto Universitario de Materiales, Universidad de Alicante, Ap. 99, E-03080 Alicante, Spain; jhony.flores@ua.es (J.X.F.-L.); morallon@ua.es (E.M.); 2Department Química Inorgánica e Instituto Universitario de Materiales, Universidad de Alicante, Ap. 99, E-03080 Alicante, Spain

**Keywords:** carbon nanotubes, carbon defects, edge sites, topological defects, ORR

## Abstract

Defect-containing carbon nanotube materials were prepared by subjecting two commercial multiwalled carbon nanotubes (MWCNTs) of different purities to purification (HCl) and oxidative conditions (HNO_3_) and further heat treatment to remove surface oxygen groups. The as-prepared carbon materials were physicochemically characterized to observe changes in their properties after the different treatments. TEM microscopy shows morphological modifications in the MWCNTs after the treatments such as broken walls and carbon defects including topological defects. This leads to both higher surface areas and active sites. The carbon defects were analysed by Raman spectroscopy, but the active surface area (ASA) and the electrochemical active surface area (EASA) values showed that not all the defects are equally active for oxygen reduction reactions (ORRs). This suggests the importance of calculating either ASA or EASA in carbon materials with different structures to determine the activity of these defects. The as-prepared defect-containing multiwalled carbon nanotubes exhibit good catalytic performance due to the formation of carbon defects active for ORR such as edge sites and topological defects. Moreover, they exhibit good stability and methanol tolerances. The as-prepared MWCNTs sample with the highest purity is a promising defective carbon material for ORR because its activity is only related to high concentrations of active carbon defects including edge sites and topological defects.

## 1. Introduction

Fossil fuel depletion has led to the development of alternative energy suppliers that could suppress their use and promote more sustainable energy sources [1]. In this field, electrochemical devices have emerged as a real alternative to meet this purpose, with fuel cells being the most promising to transform chemical energy into electrical energy [2,3]. However, the reactions necessitate catalysts because of their slow kinetics, especially the oxygen reduction reaction (ORR) whose rate is more than six orders lower than the hydrogen oxidation reaction (HOR) [4]. The commercial catalysts are based on Pt-based materials because of their great catalytic performance in both acidic and alkaline media [5]. However, these materials have drawbacks related to their high-cost and low-abundance, and they can even exhibit stability and poisoning issues [6]. Therefore, developing materials based on low-cost and abundant elements is mandatory.

In this sense, many authors have developed potential material alternatives to platinum for ORR such as transition metal oxide-based materials [7], noble metal alloys [8], transition metal-based nitride materials [9], N4 metal macrocyclic complexes [10], and carbon-based materials [11]. Among them, metal-free carbon materials are highlighted due to their outstanding catalytic response mainly in alkaline mediums, as well as their good electrical conductivity, high surface area, excellent thermal stability, and high chemical and catalytic stability [12,13], which could lower the price for large-scale implementation. Generally, metal-free sp^2^ carbon materials doped by heteroatoms (N, P, S, B) have shown great catalytic activity by breaking the integrity of π conjugation, which induces the generation of charged sites that are favourable for O_2_ adsorption, and subsequent O-O bond breaking [14,15,16]. However, Zhao et al. [17] suggested that the incorporation of heteroatoms is not responsible for catalytic activity, but the defects generated in the carbon matrix by their presence.

Previous studies reported that a change in the charge distribution can be produced also by the presence of carbon defects and not only by doping with heteroatoms [18]. Hence, carbon defects can have the same effect on electron distribution, breaking the integrity of π conjugation and promoting ORR activity [19,20]. Among the different carbon defects, edge sites were reported to be important active sites for ORR [21,22,23]. The importance of edge sites in ORRs was reported by Shen et al. [21] by studying ORRs on highly oriented pyrolytic graphite positioned on edge or basal planes in ORR, revealing the higher catalytic activity of edge sites. DFT calculations performed by Zhong et al. [24] revealed that the enhancement in ORR activity originated from the electron transfer between edge carbon atoms and the carbon atoms around the edge, which favours the O_2_ adsorption. Among zigzag and armchair edge defects, Jiang et al. [19] revealed through theoretical calculations that zigzag edge defects contribute to increased ORR activity. Jiang et al. [25] attributed this effect to the fact that zigzag edge sites can induce extra spin density that increases active site density and intrinsic activity for ORR. Radovic et al. [26,27] suggested that zigzag edge sites are carbene-like, which can be active sites for O_2_ adsorption.

In addition to edge site defects, topological defects have been also reported to be very active for ORR. Topological defects are typically created by the removal of heteroatoms existing in the carbon materials by thermal treatments [19,28,29,30]. These heteroatoms can be introduced in carbon materials during their synthesis or afterwards by a doping procedure. Zhao et al. [28] reported the highly active G585 defect (pentagon-octagon-pentagon) generated by nitrogen removal. The activity of this defect was comparable to Pt for ORR under DFT calculations. On the contrary, Wang et al. [29] reported that the C5 defect (pentagon) is preferable for ORR, while the G585 defect is more favourable for hydrogen evolution reaction (HER). Moreover, it was observed by theoretical studies that both C5 and zigzag edge sites contribute positively to increased ORR activity [19].

Another method for creating carbon defects is through the removal of metals existing in carbon materials [22,31]. Zhong et al. [22] reported that chemical drilling of carbon nanotubes by supporting cobalt results in a carbon material with good catalytic activity due to a high concentration of carbon defects and edge sites. It was also revealed that an optimum cobalt concentration is necessary because an excess increases the electron transfer resistance, decreasing the ORR performance. Interestingly, the removal of metal traces remaining in carbon nanotubes from their synthesis with acids and then subjecting them to a thermal treatment at high temperature also leads to active carbon defects [31]. Waki et al. [31] ascribed good catalytic activity to the formation of topological defects by the removal of CO at high temperature. The carbon reconstruction at high temperature would lead to the formation of C5+7 defects (pentagon-heptagon) responsible for the enhancement in ORR activity. Even Tang et al. [32] observed that C5+7 defects have a higher activity than C5 by offering optimal adsorption of oxygen intermediates.

The presence of defects in carbon materials is commonly determined using Raman spectroscopy, particularly by assessing the value of the I_D_/I_G_ ratio. This method is frequently used to characterize carbon materials thermally treated or doped with heteroatoms and/or metals, whose carbon structure is mostly similar to the pristine one [12,18,19,30]. This might facilitate the identification of the active site defects in ORR. On the contrary, this technique is not very useful to characterize carbon materials with very different structures since it provides average structural information. In this sense, other more specific and selective parameters such as the reactivity and the active surface area (ASA) could provide important information about the catalytic active sites. Gabe et al. [33] outlined a correlation between both the ASA and O_2_ reactivity in the gas phase with the ORR catalytic activity of carbon materials with different structures. Thereby, the combination of these parameters along with Raman spectroscopy could help identify the catalytic active sites present in carbon materials towards ORR.

In this work, we propose the development of defective carbon materials through a thermal treatment of commercial multi-walled carbon nanotubes (MWCNTs) that were previously oxidized with nitric acid. The as-synthesized carbon nanotube materials were characterized by different physicochemical and electrochemical techniques. It was observed that the combination of both experimental procedures was essential to generate edge sites and topological defects that are active for ORR. One of the materials prepared is a promising defective carbon material with a good catalytic activity which is only related to the high concentration of active carbon defect sites. Moreover, it shows acceptable stability in ORR. Moreover, carbon materials with different structures including the as-prepared MWCNT-based materials were characterized by Raman spectroscopy to quantify the number of carbon defects and how these could promote the ORR. In this sense, the determination of the ASA and electrochemical active surface area (EASA) values was important to relate the catalytic activity of the carbon materials with the nature of their carbon defects. Meanwhile, Raman spectroscopy is useful only for carbon materials with the same structures.

## 2. Experimental

### 2.1. Materials and Reagents

The reagents used were two different types of multi-walled carbon nanotubes: one with >99% purity, an outer diameter of 13–18 nm, and a length of 3–30 µm (Cheap Tubes Inc., Brattleboro, VT, USA), and another with >95% of purity, an average diameter of 14 nm, and lengths ranging from few hundred nanometers to several microns (Nanoblack, Columbian Chemicals Co., Ltd., Marietta, GA, USA). Potassium hydroxide (KOH) (VWR Chemicals, Prague, Czech Republic), hydrochloric acid 37% (PanReac AppliChem ITW Reagents, Darmstadt, Germany), nitric acid 65% (PanReac AppliChem ITW Reagents, Darmstadt, Germany), 2-propanol ACS (Reag., Darmstadt, Germany), Nafion^®^ 5% *w*/*w* water and 1-propanol (Alfa Aesar, Kandel, Germany), and 20 wt% Pt/Vulcan (Sigma-Aldrich, St. Louis, MO, USA). The solutions were prepared in ultrapure water (18 MΩ/cm from a Millipore^®^ Milli-Q^®^ water system). The gases, N_2_ (99.999%), O_2_ (99.995%), H_2_ (99.999%), and synthetic air were provided by Air Liquide, and they were used without any further treatment.

Apart from the above-mentioned carbon nanotube materials, the following carbon materials were also used: the XC-72F Vulcan carbon black (XC72) (Cabot Corporation, Boston, MA, USA), a highly microporous activated carbon (YPF) (Kuraray, Okayama, Japan), single-walled/double-walled carbon nanotubes (SW) with 99% purity (outer diameter of 1–2 nm and length of 3–30 µm) (Cheap Tubes Inc., Brattleboro, VT, USA), a microporous char (AC), and carbon nanofiber (CNF). These carbon materials are the same used by Gabe et al. [33] in their research.

### 2.2. Purification and Thermal Treatment of the MWCNTs

To perform this study, MWCNTs were selected because these materials could lead to a higher number of carbon defects related to the broken walls. This assumption was made in relation to the study performed by Gabe et al. [33]. In this work, single-walled carbon nanotubes (SWCNTs) developed carbon defects after oxidative treatment with nitric acid and a subsequent thermal treatment. Moreover, as in the synthesis of multi-walled carbon nanotubes, transition metal-based catalysts are employed. Thus, the final carbon products can have some metal impurities which can be active sites for ORR and can hide the real activity of the metal-free electrocatalysts [34] making a detailed purification process necessary. The removal of the metal traces might be similar to a chemical drilling method that leads to the formation of carbon defects [22]. Thereby, the materials have been purified following the purification treatment described in the literature [33]. This consists of mixing 200 mg of the carbon material with 100 mL of 5 M HCl at 50 °C overnight under reflux conditions. Then, the sample was washed with distilled water until pH 7 and then they were dried at 110 °C on the stove.

The as-obtained carbon materials were subjected to oxidation treatment employing the same ratio of weight and acid, but in this case with 3 M HNO_3_ at 120 °C for 24 h under reflux conditions, similar to the procedure described elsewhere [35]. Later, the materials were washed until the pH was neutral and dried at 110 °C on the stove. Finally, the carbon materials were heat-treated up to 920 °C in the N_2_-atmosphere at a heating rate of 5 °C min^−1^ with a flow rate of 100 mL min^−1^ and kept for 30 min. The materials were labelled as MW99_P and MW95_P for pristine carbon materials, whereas MW99_T and MW95_T were for the as-prepared materials. Appendix A detailed the experimental procedure employed for the synthesis of the as-prepared MWCNTs.

### 2.3. Physicochemical Characterization

The surface composition of the materials was characterized by X-ray photoelectron spectroscopy (XPS) in a VG-Microtech Multilab 3000 equipment with an Al Kα radiation source (1253.6 eV). The morphologies of the materials were analysed by transmission electron microscopy (TEM, JEOL-2010, 200 kV accelerating voltage, Akishima, Japan). The TEM microscope is equipped with an INCA Energy TEM 100 model X-ray detector and a GATAN acquisition camera. Energy-dispersive X-ray spectroscopy (EDX) was used to characterize the bulk composition of materials.

The carbon materials were also analysed using the Thermo Finnigan Flash 1112 elemental microanalyzer for elemental analysis.

The surface area of the carbon materials was analysed by physical adsorption of N_2_ (−196 °C) using an automatic adsorption system Autosorb-6 and an Autosorb Degasser from Quantachrome Instruments (Boynton Beach, FL, USA). Before the nitrogen adsorption, the samples were outgassed at 250 °C under vacuum for 8 h. The Brunauer–Emmett–Teller (BET) surface area values were calculated from the obtained isotherms.

Temperature-Programmed Desorption (TPD) measurements were employed to characterize the surface chemistry of carbon materials. A TGA-DSC equipment (Simultaneous TGA/DSC SDT Q600, TA Instruments, New Castle, DE, USA) coupled to a mass spectrometer (HiCube 80 Eco, Pfeiffer Vacuum, Aßlar, Germany) was used to follow the CO and CO_2_ desorbed species from the decomposition of surface functional groups.

Raman spectra were obtained using a Jasco NRS5100 spectrometer with a 3.9 mW solid-state laser (green) at 532 nm, a 20× MPLFLN objective, and a focal distance of 300 nm. Each spectrum was acquired for 120 s. Calibration of the spectrometer was performed with a Si slice (521 ± 2 cm^−1^).

### 2.4. Carbon-Oxygen Gasification Characterization

The active surface area (ASA) and the reactivity of the materials can be good descriptors to evaluate the electroactivity of the carbon materials. The ASA of the carbon materials was measured employing the following procedure described elsewhere [33]. Firstly, approximately 10 mg of the carbon materials were heated up to 920 °C at a heating rate of 20 °C min^−1^ with a flow rate of 100 mL min^−1^ and kept at 920 °C for 30 min under N_2_ to remove oxygen complexes on the carbon surface. Then, the temperature is lowered to 250 °C and kept for 1 h, maintaining the N_2_ atmosphere. Next, synthetic dry air (20 vol% O_2_ in N_2_) is introduced in the thermobalance for 7 h to produce the oxygen chemisorption. Finally, the thermobalance is fed with N_2_ for 90 min. The ASA was determined from the weight uptake of the carbon materials by the following equation considering that the chemisorbed oxygen atom occupies an area of 0.083 nm^2^ [36].
(1)$$ASA=\frac{1}{w_{0}}\cdot \frac{A\cdot n\cdot \left(w_{c}-w_{0}\right)}{N_{0}}$$
where *A* is the area that one oxygen atom occupies per carbon atom, *n* is the Avogadro number, *w*_0_ is the weight of the material before the chemisorption step (inert atmosphere), *w_c_* is the weight of the carbon after the oxygen chemisorption (inert atmosphere), and *N*_0_ is the oxygen atomic weight.

Reactivities of the materials were determined by isothermal TGA analysis under synthetic dry air with a procedure similar to the one described elsewhere [33]. Approximately 2 mg of the carbon materials were heated in a He atmosphere up to 920 °C at a heating rate of 20 °C min^−1^ and kept for 30 min. Then, the temperature is decreased to 550 °C and maintained for 60 min. Finally, the atmosphere is changed to synthetic dry air and maintained for 60 min, while monitoring the weight changes over time. The reactivity at this temperature was calculated according to the following equation [37]:(2)$$R_{550}=-\frac{1}{w_{0}}\cdot \frac{dw}{dt}$$
where *R*_550_ is the initial reactivity (gg^−1^ h^−1^), *w*_0_ is the initial mass of the sample (g) before the introduction of synthetic dry air, and *dw*/*dt* is the initial rectilinear weight loss rate (gh^−1^). The TGA/DSC SDT Q600 equipment with a sensitivity of 1 µg was used in the measurements.

### 2.5. Electrochemical Characterization

In order to electrochemically characterize the samples, an ink was prepared with a concentration of 1 mg mL^−1^. This was achieved by sonicating 1 mg of the carbon material with 1 mL of a solution containing 20 vol% 2-propanol and 0.02 vol% Nafion^®^ in water.

The electrochemical measurements were performed at 25 °C in a three-electrode cell in 0.1 M KOH solution employing an Autolab PGSTAT302 potentiostat (Metrohm, The Netherlands). The working electrode used for the measurements was a rotating ring-disk electrode (RRDE) from Pine Research Instruments (Durham, NC, USA) which consisted of a glassy carbon (GC) disk (5.61 mm diameter) and an attached Pt ring. A graphite bar was employed as a counter electrode, while a reversible hydrogen electrode (RHE) immersed in the lugging capillary was used for the reference electrode.

Cyclic voltammetry (CV) and linear sweep voltammetry (LSV) experiments were performed for the electrochemical characterization. The loading of electroactive materials deposited on the glassy carbon was 480 µg cm^−2^. Prior to conducting the cyclic voltammetry (CVs), the cell was purged with nitrogen for 20 min, and the CVs were performed with potential limits ranging from 0 to 1 V (vs. RHE) at a scan rate of 50 mV s^−1^.

LSV experiments were performed to study the electroactivity of the materials in ORR employing different rotation rates between 400 and 2025 rpm at 5 mV s^−1^ from 1 to 0 V (vs RHE) in an O_2_-saturated solution. The Pt ring electrode was kept at 1.5 V during all the measurements. The electron transfer number (n_e−_) was calculated according to the following equations, which relate the oxidation of the hydrogen peroxide at the Pt ring electrode with this value [38]:(3)$${HO}_{2}^{-}\left[\%\right]=200\times \frac{I_{ring}/N}{I_{disk}+I_{ring}/N}$$
(4)$$n_{e^{-}}=\frac{4I_{disk}}{I_{disk}+I_{ring}/N}$$
where I_disk_ and I_ring_ are the currents, in absolute values, measured at disk and ring, respectively, and N is the collection efficiency of the ring, which was experimentally determined as 0.37.

The electrochemical active surface area (EASA) of the as-prepared materials was obtained by performing CVs at different scan rates. To accomplish this, 30 µg of the ink previously described was deposited on a glassy carbon of 3 mm diameter and CVs were performed at 2–400 mV s^−1^ in a 0.1 M phosphate buffer solution (PBS, pH = 7.2) in the presence of the redox probes 10 mM K_3_Fe(CN)_6_/10 mM K_4_Fe(CN)_6_. These experiments show how redox peaks associated with Fe^2+^/^3+^ redox couple behave at different scan rates, in terms of intensity and separation of the redox peaks. This information is useful in determining the EASA and heterogeneous electron transfer rate constant (kº) for this redox reaction. The methods employed for calculating these parameters will be discussed in detail later in the manuscript.

## 3. Results and Discussion

### 3.1. TEM Analysis of the Materials

Nanostructure characterization is essential to observe any possible changes in the MWCNTs after the acidic and thermal treatments, which could affect their electrocatalytic activity. Thus, the materials were analysed by TEM and the images are depicted in Figure 1. It can be distinguished that pristine materials MW99_P (Figure 1a,b) and MW95_P (Figure 1e,f) contain metal nanoparticle traces related to the catalysts used for carbon nanotube synthesis [39]. Moreover, the concentration of metal nanoparticles seems to be related to the purity of the carbon nanotubes, and this is more notorious in the sample MW95_P which has a lower MWCNTs purity (Figure 1e). It can detect the presence of defects related to disordered carbon and hole defects in both pristine materials (shown by white arrows in Figure 1b,f). The complete removal of metal nanoparticles can be observed in the MW99_T sample resulting from the HCl treatment (Figure 1c,d). However, some metal impurities can still be distinguished in the inner nanotube walls of the MW95_T sample, which could influence the ORR catalytic activity (Figure 1g). Although the as-prepared materials were subjected to a high-temperature treatment, disordered carbon structures were observed, which can contain active sites (Figure 1d) [30]. In addition, other defects can also be distinguished including lattice fringes, broken walls, and holes (Figure 1d,h) which could increase the amount of pentagon and edge site defects that are active for ORR [19].

### 3.2. Characterization of the Surface Chemistry and Bulk Composition

It is well-known that the surface composition of the materials can differ significantly from the bulk composition. Therefore, it is important to characterize both regions separately employing different techniques. The bulk composition of the materials was characterized by two different techniques. The EDX confirms the presence of metal traces in pristine materials such as Mo, Co, Mg, and Mn employed for the synthesis of carbon nanotubes, as shown in Table 1 [39,40]. Meanwhile, the treated samples show a decrease in metal traces, especially the sample MW99_P. This fact confirms the crucial role of HCl in the elimination of metal impurities [41]. However, the MW95_P sample still contains metal traces, but their concentration was reduced by approximately half after treatment. This confirms the difficulty of eliminating the metal from the inner part of the nanotubes. This agrees with the TEM results previously discussed and with those obtained by TG experiments (Appendix A). Moreover, it is observed that oxygen content reduces considerably after thermal treatment, i.e., almost all the oxygen-containing functional groups were eliminated. Elemental analysis technique was used to determine the concentration of C, H, N, and S in the carbon nanotube materials. Only C and H were detected, and the results in Table 1 show the increase in the C content after the applied treatments. The results of elemental analysis agree with the TG measurements (Appendix A).

In contrast, the XPS technique can provide information about the surface concentration and the nature of the different species. Appendix A shows the XPS spectra for all the materials revealing mainly the presence of C 1s and O 1s emissions. Concerning the C 1s spectra of the MWCNT-based samples, no differences can be observed among them, indicating the presence of the same C species on the surface of all materials (Appendix A). In the case of the O 1s spectra, all samples, except for the MW_95P, have only one peak at the energy value close to 532.6 eV, which is indicative of oxygen functional groups such as carbonyl and carboxylic groups (Appendix A) [42]. The MW_95P sample has another peak at the binding energy value close to 530.2 eV that might be attributed to lattice oxygen from metal oxides. The possible metal oxides formed from the metal traces (detected by EDX) can be MoO_3_ [43] and Co_2_O_3_ [44]. Moreover, low intensity of the N 1s signal was detected for the pristine carbon nanotube materials (Appendix A). Any signal related to the metal nanoparticles observed by TEM was detected, indicating that they are mainly confined inside the carbon nanotubes and their concentration on the surface is low. Table 1 summarizes the data obtained from XPS analysis. The reduction in oxygen and nitrogen species in samples MW99_T and MW95_T confirms that the thermal treatment enriches the carbon content on the surface to the detriment of heteroatoms. Therefore, the removal of heteroatoms may lead to the formation of defect sites. Previous studies reported that N removal can lead to the formation of topological defects including G585 and C5 which are active for ORR [28,29,30]. Among these two defects, it was determined that C5 defects have a higher intrinsic activity in ORR than G585 defects [29].

Surface oxygen functionalities of the carbon materials were characterized by temperature-programmed desorption (TPD). This technique is widely used in the characterization of carbon materials because it provides quantitative information on individual surface functional groups [45]. The oxygen-containing functional groups decompose mainly as CO and CO_2_. Figure 2 displays the CO and CO_2_ gas evolution profiles during the TPD for the carbon materials. In all materials, the CO evolution is larger than the CO_2_ one. Regarding the evolution temperatures, the oxygen-containing functional groups desorbing as CO in the pristine materials might be related to phenol, carbonyl, and quinone groups [46,47]. Meanwhile, the concentration of groups that desorb as CO_2_ is insignificant, but they can be associated with lactone group decomposition according to the desorption temperature [48]. The sharp peak at around 800 °C in the MW95_P sample indicates the presence of metal traces. As a consequence of the HCl treatment, the sharp peak disappears, showing that the metal impurities were almost completely removed [33,48].

Table 2 shows the quantification of CO and CO_2_ evolved during the experiments for all samples. The total oxygen content is estimated from the sum of CO and 2CO_2_. It can be observed that the thermal treatment reduces considerably the amount of oxygen-containing functional surface groups. The remaining oxygen groups desorb mainly as CO and might be related to carbonyl groups [46,47]. For the MW95_T sample, some of the desorbed CO might also be due to metal traces.

It is known that the removal of functional groups during the thermal treatment through CO and CO_2_ evolution leads to the generation of edge plane and topological defects in the carbon matrix. It was suggested that the removal of CO at high temperatures generates topological defects such as C5+7 defects from the reconstruction of the carbon lattice [31] and it was determined that C5+7 defects have a slightly higher catalytic activity than C5 defects showing optimal adsorption of oxygen intermediates [32].

### 3.3. Porous Texture and Carbon-Oxygen Gasification Properties

Appropriate porous texture is essential for good electrochemical performance because it can facilitate the diffusion of the electrolyte and reactants through the material, along with the release of products. Moreover, the number of active sites available can increase enhancing the overall electrochemical reactions. In this study, N_2_ isotherms were accomplished for all carbon materials and the BET surface area results are shown in Table 2. The BET surface area of the treated materials increases considerably, being almost 1.5 and 1.8 times higher than the pristine samples for MW99_T and MW95_T, respectively. The increase in the BET surface area agrees with the results observed by TEM, where the break of nanotube walls generates additional adsorption sites and allows the inner side of the nanotube to be accessible. Therefore, the HNO_3_ treatment followed by the thermal treatment is important to increase the BET surface area, which could lead to an enhancement in ORR activity.

The active surface area and the reactivity of the carbon materials can be good descriptors to evaluate the electroactivity of the carbon materials as has been previously demonstrated [33]. ASA provides information on the number of reactive carbon atoms or carbon active sites which might be related to edge planes such as stacking faults, single and multiple atom vacancies, and dislocations [49]. Therefore, carbon materials prepared after the thermal treatment are expected to have a higher ASA value resulting from the increase in defects. Interestingly, sample MW99_T shows a high ASA value compared to the pristine material, which proves the higher content of carbon defects which can be active for dissociative oxygen chemisorption [50]. However, the metal traces could lead to the miscalculation of the ASA by reacting with the O_2_, especially for active metals for ORR such as Co [22]. This effect can clearly be observed in the MW_95T sample which shows a decrease in the ASA value compared to the pristine material MW_95P due to the lower presence of metal traces. However, when comparing the two treated materials MW_95T and MW_99T, a decrease in ASA is clearly observed due to the lower concentration of disordered carbon in the highest purity sample. This observation is also in agreement with the measured reactivity in oxygen (see R_550_ in Table 2) (Appendix A contains the thermogravimetric profiles). It must be noted that the reactivity values for the pristine materials are strongly determined by the metal species that catalyse carbon gasification.

### 3.4. RAMAN Spectroscopy

Raman spectroscopy is an important technique to characterize carbon materials and provides information about structural disorder or further functionalization [51]. Figure 3 shows the Raman spectra of the different carbon materials observing some differences between the pristine and treated materials. Firstly, in the first-order region, two strong bands can be distinguished at around 1350 cm^−1^ and 1580 cm^−1^, corresponding to the D and G bands, respectively [51]. The D band is related to the defects or disorder existing on the carbon material including grain boundaries, vacancies, pentagons, heptagons, and edge sites [51], whereas the G band is related to the graphitic sp^2^ carbon. In the second-order region, three bands can be observed at around 2690 cm^−1^, 2940 cm^−1^, and 3230 cm^−1^ associated with 2D (or also named as G′), D + D′, and 2D′, respectively [52,53]. Among these bands, the presence of defects does not influence the appearance of the 2D and 2D′ bands, but they are required for the activation of the D + D′ band [54]. Previous studies reported a good correlation between the bandwidth of the D + D′ band and the reactivity of carbon materials [55]. Therefore, the second-order region was deconvoluted with Lorentzian functions (Appendix A) and the value of bandwidth for the D + D′ band is shown in Table 3. It can be observed that, compared to their corresponding pristine materials, there is a decrease and increase in bandwidth for MW99_T and MW_95T samples, respectively, confirming the relationship between the reactivity and structure of the MWCNTs.

Surprisingly, the MW95-based samples exhibit a lower intensity for the D band related to the G band in comparison to the MW99-based samples. This suggests that not all the defects present in the carbon nanotubes can be active surface sites according to the different tendencies observed by ASA values. Moreover, the smaller diameter of the MW95-based materials can increase the curvature of the carbon nanotubes resulting in a higher number of active sites [56].

To better analyse the nature of the defects and perform an adequate analysis, the D and G bands were deconvoluted into Lorentzian functions and the results are depicted in Figure 4. The MW95-based materials are well-fitted with the D and G bands, whereas the MW99-based materials present three additional bands [52]. The D* band at 1184 cm^−1^ is associated with disordered graphitic lattices provided by sp^2^-sp^3^ bonds. At 1480 cm^−1^ the D″ band appears which is related to amorphous phases of carbon materials. The presence of Csp^3^ and structural defects are responsible for the activation of the D′ band at 1606 cm^−1^. Therefore, these three bands indicate the high concentration of carbon defects in MW99-based materials.

To quantify the number of carbon defects in the as-prepared materials, the I_D_/I_G_ intensity ratio was calculated (Table 3). As can be observed, the treated materials show larger amounts of defects. In the case of the MW95_T, it almost doubles the concentration compared to the pristine material. The origin of these defects might be related mainly to the increase in edge plane defects through the break of walls and fridges from carbon nanotubes. However, part of these defects might also be related to topological defects generated by the removal of heteroatoms such as N and O from the sp^2^ carbon matrix as was previously described.

Previous studies [57] reported that the nature of the carbon defects can be identified by the I_D_/I_D′_ intensity ratio. According to this model, a value of 3.5 is related to boundary-like defects, 7 corresponds to vacancy-like defects, 10 is associated with hopping defects, and 13 is attributed to the presence of sp^3^-related defects. As the D′ band was only detected for the MW99-based samples, this model was applied just for these samples. The value of 3.61 indicates that these samples mainly contain boundary-like defects such as alignment of 5-8-5 defects (pentagon-octagon-pentagon) or alternation of pentagons and heptagons [58]. These defect sites can contribute positively to the catalytic activity in ORR. However, it is desirable to have these topological defects isolated over one plane and not between two planes.

Another ratio providing information about the morphology of the carbon nanotubes is I_2D_/I_D_, which was demonstrated to have a linear dependence on the diameter of the MWCNTs [59]. For the treated carbon materials, this value decreases corroborating that the diameter of the CNTs is affected by the chemical treatment with HNO_3_ and further heat treatment which breaks the nanotube walls and lattice fridges.

The I_D_/I_G_ ratio can be employed to characterize some parameters of the defects such as the estimation of the in-plane size of sp^2^ domains (L_sp2_), average defect distance (L_D_), and defect density (n_D_, cm^−2^) using the following equations [54,60]:(5)$$L_{{sp}^{2}}=\frac{560}{E_{L}^{4}}\frac{I_{G}}{I_{D}}$$
(6)$$L_{D}^{2}\left(nm^{2}\right)=2.4\times 10^{-9}\lambda_{L}^{4}\frac{L_{G}}{L_{D}}$$
(7)$$n_{D}=\frac{2.4\times 10^{22}}{\lambda_{L}^{4}}\frac{L_{D}}{L_{G}}$$
where *E_L_* represents the energy of the Raman laser source with a value of 2.33 eV and *λ_L_* is its wavelength (532 nm).

It can be observed that the pristine carbon nanotube materials have a higher L_sp_^2^ value. Generally, the ASA value is inversely related to the L_sp_^2^ value, which is in agreement with the MW_99-based samples. However, for the MW95-based materials, both values decrease for the as-prepared materials which confirms that the high ASA value for these samples, and especially for MW95_P, is influenced by metal traces. As expected, the average defect distance decreases for the treated carbon nanotubes. In summary, there is an increase in carbon defects in each series of the treated materials related to topological and edge site defects that would strongly influence the ORR activity.

### 3.5. Electrochemical Characterization

The physicochemical characterization of the carbon nanotubes revealed important differences resulting from the generation of carbon defects. Therefore, it is expected to have some notorious differences in the electrochemical properties. Firstly, it will be examined the electrochemical behaviour of electron transfer at the electrode/electrolyte interface for all carbon materials by using a redox probe. The redox probe selected for this purpose is the Fe(CN)_6_^3−/4−^ couple, which is classified as surface-sensitive but not as an oxide-sensitive redox system [61]. In carbon nanotube materials, this redox couple is facilitated by the presence of edge planes exposed to the solution. Appendix A displays the different CVs performed at different scan rates in the presence of the redox probe for all the carbon materials. From the peak separation, it is possible to calculate the heterogeneous electron transfer rate constant (k°) using the Nicholson method [62].

Thus, considering an average value of the diffusion coefficient between the oxidized and reduced iron species (*D*), and a balanced anodic and cathodic charge transfer coefficient (α = 0.5), *k^0^* can be estimated according to the following equation:(8)$$k^{0}={\left(\pi \frac{DnF}{RT}v\right)}^{1/2}\psi$$
where *n* is the number of electrons transferred, *F* is the Faraday constant, *R* is the gas constant, *T* is the temperature, and *ψ* is a dimensionless charge transfer parameter that depends on the peak potential separation (ΔEp). Depending on the ΔEp value, the *ψ* can be found tabulated in the original Nicholson’s paper for the range 61–212 mV. However, it can be easily obtained using the mathematical expression proposed by Lavagnini et al. [63]. Meanwhile, for strong irreversible systems (ΔEp > 212 mV), the numerical approach developed by Mahé et al. is required [64]. From plotting *ψ* vs. ${\left(\frac{\pi DnF}{RT}\right)}^{-1/2}v^{-1/2}$, the *k*^0^ value can be obtained (Table 4). The higher value of *k*^0^ for the treated carbon nanotube materials proves that an increase in edge plane concentration favours the charge transfer for carbon nanotubes [65].

The EASA of the carbon nanotube materials, which provides information about the active surface area available for the electron transfer to species in solution, was calculated according to the Randles–Sevcik equation [61]. The equation relates the peak current for an electron transfer-controlled process with the root of the scan rate, as it is shown:(9)$$I_{p}=2.687\times 10^{5}ACn^{3/2}(Dv{)}^{1/2}$$
where *I_p_* is the peak current (A), *A* is the electroactive area (cm^2^), *C* is the concentration of the electroactive species (mol cm^−3^), *n* is the number of exchanged electrons, *D* is the diffusion coefficient for the redox probe (cm^2^ s^−1^), and v is the scan rate (V s^−1^). The value of the diffusion coefficient used for both Fe(CN)_6_^3−/4−^ was 7.6 × 10^−6^ cm^2^ s^−1^ obtained from the literature [66]. Figure 5 shows that the anodic (I_a_) and cathodic (I_c_) peak currents vary linearly with the square root of the scan rate revealing that the mass transport of redox species onto the electrode surface is unequivocally controlled by diffusion. The electrochemical active surface area values from the anodic and cathodic peaks were quite similar, thus, an average value was calculated from A values of EASA per gram of material (Table 4). As expected, the treated materials exhibit a higher EASA because of the oxidative and thermal treatments, even the MW99_T sample doubles the value of the pristine sample. Now, comparing these values to those from ASA (Table 2), it can be observed that these are quite similar, especially for the MW99_T sample. For the MW95_P sample, a huge difference can be noticed, which might be related to the metal species that interfere with the ASA determination. Therefore, the removal of metal traces is necessary for obtaining a good correlation between both values. The ASA and EASA values are quite similar in the as-prepared materials. This confirms that most of the active sites deduced from the ASA value might be ascribed to edge sites, which are relevant for ORR activity. It is important to highlight that not all edge site defects are equally active for ORR, zigzag edge sites were reported to be more active than armchair edge sites [19]. This is related to the faster electron transfer and stronger adsorption energy towards oxygen molecules [24]. Thereby, a high concentration of zigzag edge sites can potentially increase the catalytic activity.

Once some parameters were determined by the redox probe, the carbon nanotube materials were characterized by cyclic voltammetry in N_2_ and O_2_ atmospheres in 0.1M KOH and the results are depicted in Figure 6. The treated materials exhibit a larger double-layer capacitance compared to the pristine materials, which are related to their higher BET surface area produced by the oxidation and thermal treatments. The disappearance of redox peaks in the prepared carbon nanotubes proves that the purification was successful, especially for the MW99_T sample. This is in concordance with the previous results. The anodic and cathodic peaks displayed in the MW99_P sample could be related to Mn species, whose redox processes appear at those potentials [67]. Figure 6b depicts the CVs in the O_2_ atmosphere, and it can be observed a sharp reduction peak at around 0.7–0.8 V vs. RHE, which is related to the oxygen reduction reaction. The treated carbon nanotube materials have a more positive onset potential for this process, indicating their higher electrocatalytic activity. However, the electrocatalytic activity will be discussed later by polarization curves.

### 3.6. Electrocatalytic Activity for ORR

To assess the electrochemical performance of the carbon materials in ORR, polarization curves were recorded using a rotating ring disk electrode in 0.1 M KOH medium saturated with O_2_. Figure 7a shows the liner sweep voltammograms of the carbon materials together with a commercial Pt/C material with a 20 wt% of Pt. The pristine materials have a lower catalytic activity highlighting the MW99_P sample which exhibits two reduction processes. The first reduction process is related to dioxygen reduction, while the second process corresponds to the reduction in both the remaining dioxygen and peroxide species [68]. We can observe that the treated carbon nanotube materials have an enhancement in their performance with a better onset potential and higher current density. The MW99_T sample has an even larger current density compared to the commercial Pt/C electrocatalyst. This might be related to the high specific surface area and a considerable amount of catalytic carbon defects for ORR [69]. Figure 7b shows the number of electrons transferred during the oxygen reduction reaction, revealing an increase after the different treatments. Thereby, the carbon defects generated are quite selective for a direct reduction through a four-electron pathway.

Table 4 shows the electrochemical parameters obtained for carbon nanotube materials from the LSV curves shown in Figure 7. Despite having a lower EASA value, the MW99_T sample exhibits a higher catalytic activity and selectivity than the MW95T sample. The only parameter lower is the E_1/2_ due to the greater current limiting of the MW99_T sample, which duplicates the value of the MW95_T sample. The best performance of the MW99_T sample might be related to the nature of the edge defects. Hence, the MW99_T sample seems to have a higher concentration of edge zigzag defects that are more active for ORR. Moreover, it is important to remark that MW99 T material has a higher concentration of defects as detected by Raman spectroscopy resulting from the removal of heteroatoms. Among these defects, topological defects such as C5, C5+7, and G858 might be the origin of the enhancement in catalytic activity for MW99_T material. Thus, the catalytic activity of the MW99_T sample is related to its higher concentration of active sites related to zigzag edge and topological defects.

Observing the Tafel slopes in Table 4, it can be concluded that all samples have as a rate-determining step the protonation of O^2−^ species on the active sites to form HO_2_^−^ [70,71]. The treated carbon nanotubes have lower Tafel slope values, indicating that the kinetics is faster compared to the pristine carbon nanotubes. Especially the MW99_T sample exhibits the best kinetics which, together with the large amount of catalytic carbon defects, explains its outstanding electrocatalytic performance in ORR.

The stability of the as-prepared materials was determined by a chronoamperometric study employing the RRDE at 1600 rpm in O_2_-saturated electrolyte at a constant potential of 0.65 V (Figure 8) [72]. A commercial Pt/C electrocatalyst was also tested in the same conditions for comparison purposes. After 180 min at a constant potential, methanol was added to the electrochemical cell until a 1 M concentration was reached to study the stability versus poisoning. As expected, the Pt-based electrocatalyst exhibits great stability, retaining almost 95% of the initial current after 180 min. However, after the methanol addition, the current drops immediately to zero due to the poisoning of the Pt active metal sites because of CO produced from the methanol oxidation. The activity of the treated materials decreases slowly until they stabilize at around 150–180 min, reaching a current retention of around 83 and 86% for MW99_T and MW95_T, respectively. These materials exhibit a good methanol tolerance with a minor loss of activity. The activity loss might be related to the presence of highly reactive sites that interact strongly with O_2_ molecules inactivating the sites. Despite its slightly lower stability, the MW99_T sample is considered a promising alternative to Pt-based materials due to its great catalytic activity. Moreover, the complete removal of metal traces in the MW99_T sample avoids misunderstandings about the origin of its electrocatalytic activity. Therefore, the MW99_T material is presented as a defective carbon material with good catalytic activity in ORR.

## 4. Importance of Surface Chemical Analysis and Electrochemical and Spectrochemical Results in the ORR Performance

The characterization of the MWCNT materials has shown the importance of the nature of the carbon defects in the ORR performance. Generally, carbon defects can be detected by Raman spectroscopy regardless of their activity in ORR. However, the determination of ASA can provide information about the quantity of active sites available on the carbon materials. To be precise about the nature of these active sites, the EASA using the redox probe K_3_Fe(CN)_6_ can be used for determining edge sites. Hence, it is possible to obtain descriptors to evaluate the catalytic performance of carbon materials towards ORR. In this sense, Gabe et al. [33] established a correlation between the ORR performance of carbon materials and their ASA and reactivity (R_550_) values. Therefore, these parameters are considered important descriptors to determine the ORR performance. Meanwhile, other studies observed a correlation between ASA and in-plane crystallite size (L_sp2_) in carbon nanotube-based materials determined by Raman spectroscopy [48]. The value of L_sp2_ depends on the concentration of carbon defects. However, not all carbon defects are active sites, consequently influencing the ORR activity. The present research study indicates that the EASA is similar to the ASA when most of the carbon defects are edge sites. These defects are highly active for ORR and their determination is crucial to obtain information about the ORR activity. Despite the number of topological defects not being electrochemically determined, their presence is assumed from the heteroatom removal. Interestingly, the topological carbon defects in the MW99_T sample might be responsible for its great ORR activity.

To deepen into the electrocatalytic activity of the carbon defects in the ORR, some carbon samples from Gabe et al. [33] were characterized by Raman spectroscopy and their EASA was determined. The materials have different structures such as Vulcan carbon black (XC72), activated carbon (AC), microporous activated carbon (YPF), carbon nanofibers (CNF), herringbone carbon nanotubes (herring), and single-walled carbon nanotubes (SW). The catalytic parameters obtained from Gabe et al. are shown in Appendix A together with the values of MW95_T and MW99_T samples. The as-prepared MWCNT materials have better ORR performances related to the presence of highly active sites including edge sites and topological defects.

To understand the behaviour of the carbon materials, the D and G bands were deconvoluted and the results are displayed in Appendix A, whereas the main parameters are shown in Appendix A together with the MW95_T and the MW99_T samples. Interestingly, the MW95_T sample has parameters close to SW and Herring samples. This suggests that the carbon nanotube materials develop a low number of carbon defects. Meanwhile, the activated carbon materials and the carbon black have similar parameters, except for the I_D_/I_D′_ value. In the case of the XC72 sample, this value suggests the major presence of boundary-like defects (~3.5), while the AC sample has both boundary-like defects and vacancy-like defects [57]. The YPF sample has mainly vacancy-like defects according to this parameter value (~7) [57]. The CNF sample has values similar to the MW99_T sample, so a priori its catalytic activity should be comparable, but this is not the case. This indicates the necessity to determine ASA and EASA values to evaluate properly the ORR performance.

The EASA was calculated by performing CVs at different scan rates in the presence of the redox probe (Appendix A) using the Randles–Sevcik equation. The k^0^ value was calculated by the Nicholson method. Moreover, the kinetic current density (j_k_) for ORR was calculated from the polarization curves obtained at different rotation rates using the Koutecký–Levich equation (Appendix A), as detailed in the Appendix A. These electrochemical parameters of the carbon samples are displayed in Table 5 together with the ASA, R550, and defect density (n_D_). Figure 9 and Appendix A illustrate the relationships between different parameters to obtain possible ORR descriptors. Although the MW95_T sample still contains some metal traces, it is also included in these representations. Figure 9a shows the relation between ESA and ASA, where two main correlations are distinguished. The carbon nanotube materials (blue line) have similar values of EASA and ASA, indicating that most of the defects are edge sites, and their concentration might positively influence the catalytic activity. Interestingly, the other carbon materials have a different tendency with higher ASA values than EASA values. This suggests that the carbon defects are not mainly edge sites and have another nature which seems not to be as active according to the ORR results. However, the increase in defect density (n_D_) generally favours the heterogeneous transfer electron rate constant (Figure 9b). This effect is applicable for almost all carbon materials, except for the MW95_T, which has a high k^0^ value. This implies that the metal traces influence positively on electron transfer, regardless of the defect density. Moreover, the metal traces remaining in sample MW95_T also increase the value of R_550_ as shown in Figure 9c. This explains why this sample does not follow the tendency between the j_k_ and R_550_. In the case of the MW99_T sample, despite its lower reactivity (R_550_), it exhibits the highest j_k_ due to the presence of highly active sites for ORR. The plot of EASA vs. R_550_ (Appendix A) also displayed no correlation for the MW95_T due to its higher reactivity from the presence of metal traces. Meanwhile, in the other carbon samples, a tendency can be observed between these parameters, i.e., a higher R_550_ value normally results in a higher EASA. On the contrary, the plot of k^0^ against ASA (Appendix A) leads to two different correlations depending on the ASA value. Yet again, carbon nanotube materials (blue line) and the other carbon materials (red line) have different correlations confirming the importance of the carbon structure.

From these relationships, it can be concluded that Raman spectroscopy is important to evaluate the ORR activity in carbon materials with the same nature or similar structure. In the case of carbon nanotubes, the determination of the ASA or EASA can support the Raman results, but they do not seem important for the overall discussion. On the contrary, in the other types of carbon materials including activated carbon, black carbon, and carbon nanofiber materials, the determination of ASA and EASA is compulsory to evaluate properly the ORR activity. This is because not all the carbon defects are active sites or equally active for ORR, and because the structure of the material is very important in determining the catalytic activity being especially important when materials with short or long-range order structures are compared. For example, the zigzag edge site defects are more active than the armchair ones. This mainly explains the poor activity of the YPF sample in comparison to the MW99_T sample, suggesting the higher concentration of zigzag edge site defects in the latter sample. Moreover, the MW99_T sample contains highly active topological defects.

## 5. Conclusions

Defected carbon nanotube materials were synthesized by oxidizing two different commercial multi-walled carbon nanotubes followed by a thermal treatment in an inert atmosphere. The different treatments employed reduced the amount of metal traces and oxygen-containing functional groups, especially for the MW99_T sample.

The as-prepared carbon nanotube materials exhibit a higher BET surface resulting from the break of walls of carbon nanotubes, which increases the exposed area. The presence of carbon defects was confirmed by different techniques. Raman spectroscopy revealed the increase in carbon defects on the as-prepared materials, which might be related mainly to topological and edge site defects. Topological defects might be the result of the removal of heteroatoms by the thermal treatment, whereas edge sites are formed from the generation of broken walls and fridges. Edge site defects are important catalytic sites for ORR and their concentration can be determined using the redox probe Fe(CN)_6_^3−^/Fe(CN)_6_^4−^. The MW99_T sample shows great electrocatalytic performance in ORR, which might be attributed to an increase in edge sites and topological defects such as C5 and C7+5. Regarding the nature of the carbon defects, it can be concluded that the defects are mostly related to edge sites because both EASA and ASA values are similar. Among the different edge site defects, zigzag edge sites are more active sites than armchair ones. The MW99_T sample shows good stability and methanol tolerance making it a promising alternative to Pt-based electrocatalysts.

Finally, this study suggests that Raman spectroscopy is useful for determining the catalytic activity of carbon defects only in carbon materials with similar structures. To compare carbon materials with different structures, it is necessary to calculate the ASA and EASA. These values can provide more specific information about the catalytic activity of the carbon defects determined by Raman spectroscopy because not all carbon defects appear to be active for ORR. In ORR, edge sites and topological defects are more important, and the former can be quantified by calculating EASA.

## Figures and Tables

**Figure 1 nanomaterials-14-01381-f001:**
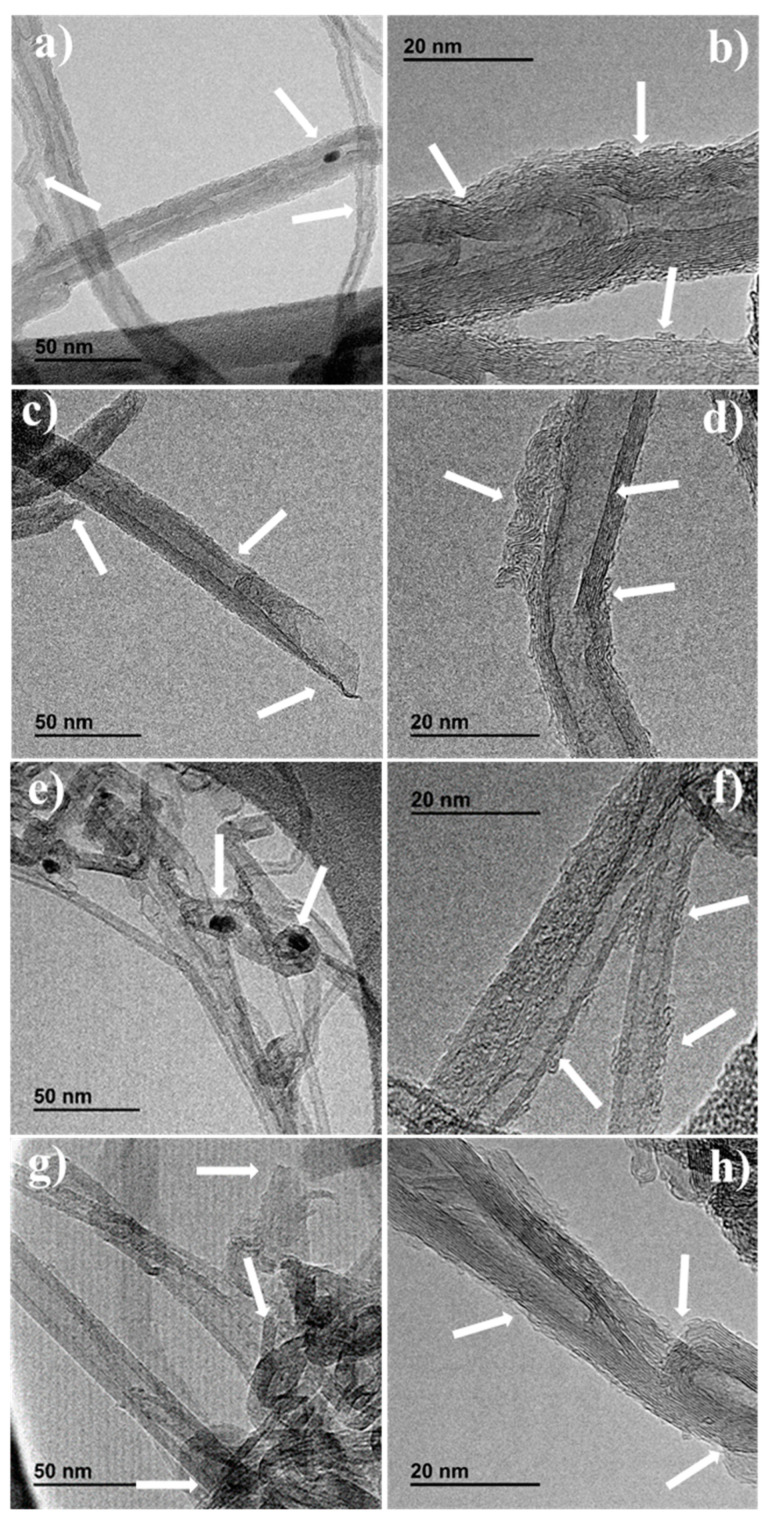
TEM images of (**a**,**b**) MW99_P, (**c**,**d**) MW99_T, (**e**,**f**) MW95_P, and (**g**,**h**) MW95_T. The arrows indicate the location of carbon defects and metal traces.

**Figure 2 nanomaterials-14-01381-f002:**
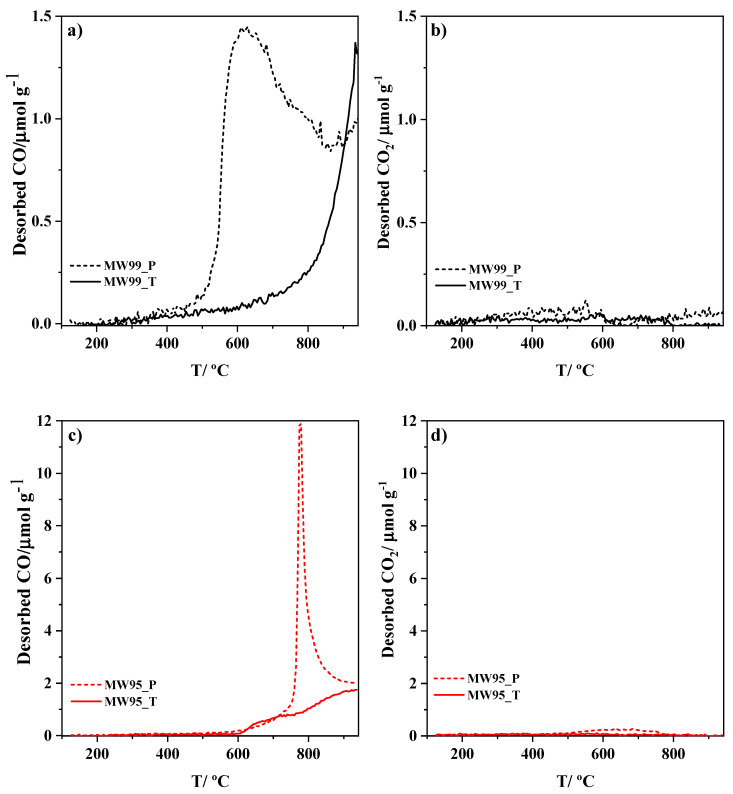
CO and CO_2_ TPD evolution profiles of the carbon materials. (**a**,**b**) MW99-based materials, and (**c**,**d**) MW95-based materials. The dashed line is for the pristine materials, whereas the solid line is for as-synthesized materials.

**Figure 3 nanomaterials-14-01381-f003:**
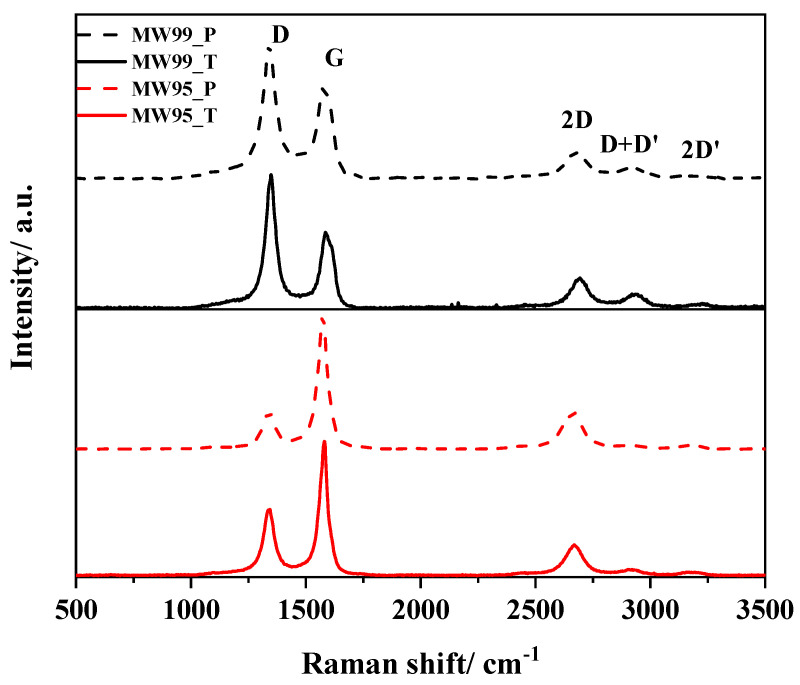
Raman spectra of the treated and pristine multi-walled carbon nanotubes.

**Figure 4 nanomaterials-14-01381-f004:**
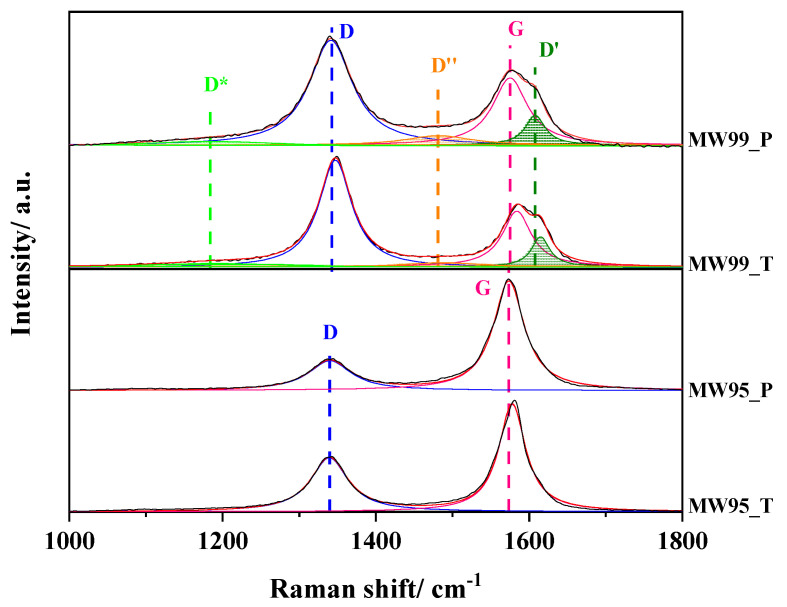
Deconvolution of the D and G bands of the treated and untreated multi-wall carbon nanotubes. The colors indicate the different bands obtained after deconvolution.

**Figure 5 nanomaterials-14-01381-f005:**
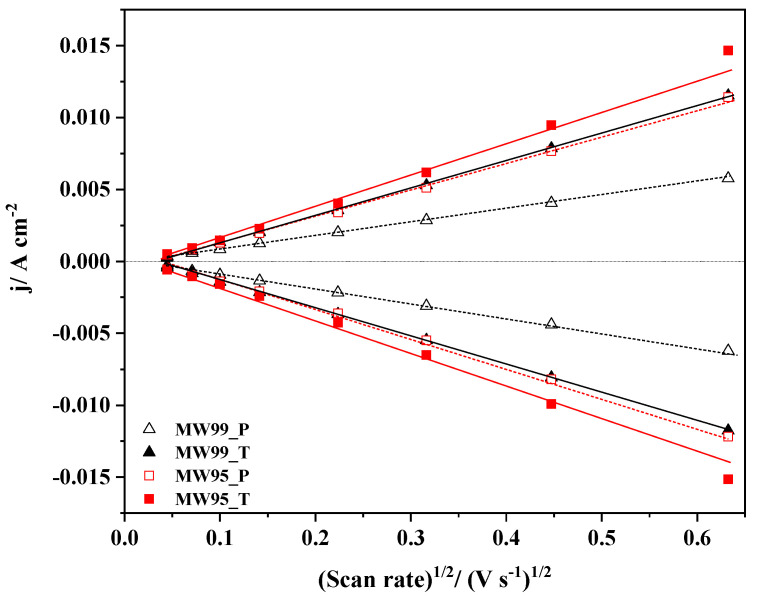
Randles–Sevcik plot obtained from the cyclic voltammetry for the carbon nanotube materials in 10 mM K_3_Fe(CN)_6_/K_4_Fe(CN)_6_ in pH 7.0 PBS solution saturated with N_2_.

**Figure 6 nanomaterials-14-01381-f006:**
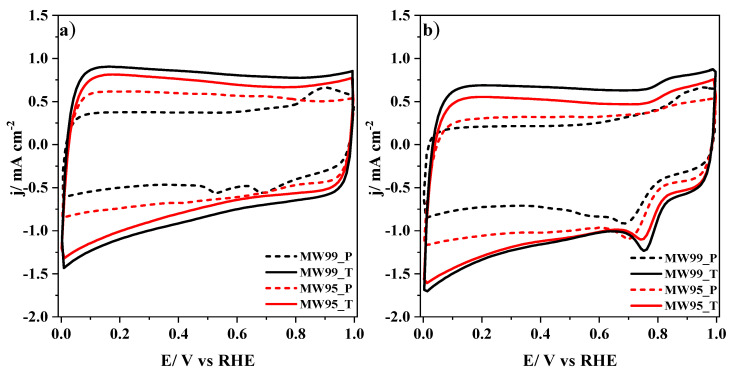
Cyclic voltammograms for carbon nanotube materials in 0.1 M KOH solution saturated with either (**a**) N_2_ or (**b**) O_2_. Scan rate: 50 mV s^−1^.

**Figure 7 nanomaterials-14-01381-f007:**
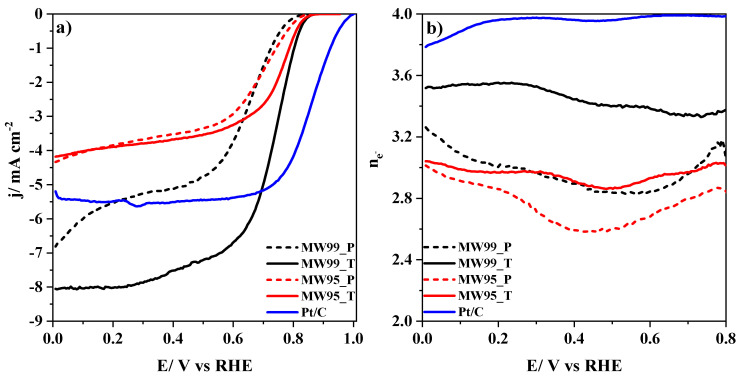
(**a**) RDE linear sweep voltammograms for carbon nanotube materials in 0.1 M KOH solution saturated with O_2_ at 1600 rpm, v = 5 mV/s. (**b**) Number of electrons transferred in ORR at increasing potential as obtained from Equation (2) by using the current measured at the ring electrode.

**Figure 8 nanomaterials-14-01381-f008:**
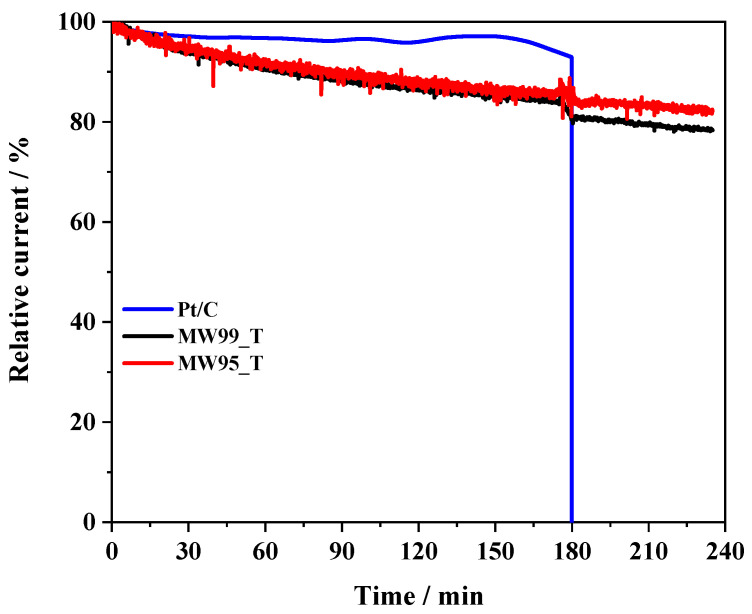
Comparative stability test for multi-walled carbon nanotube-based materials and 20 wt% Pt/C performed at 0.65 V and 1600 rpm in O_2_-saturated 0.1 M KOH and 25 °C. Methanol was added 180 min after the beginning of the experiment.

**Figure 9 nanomaterials-14-01381-f009:**
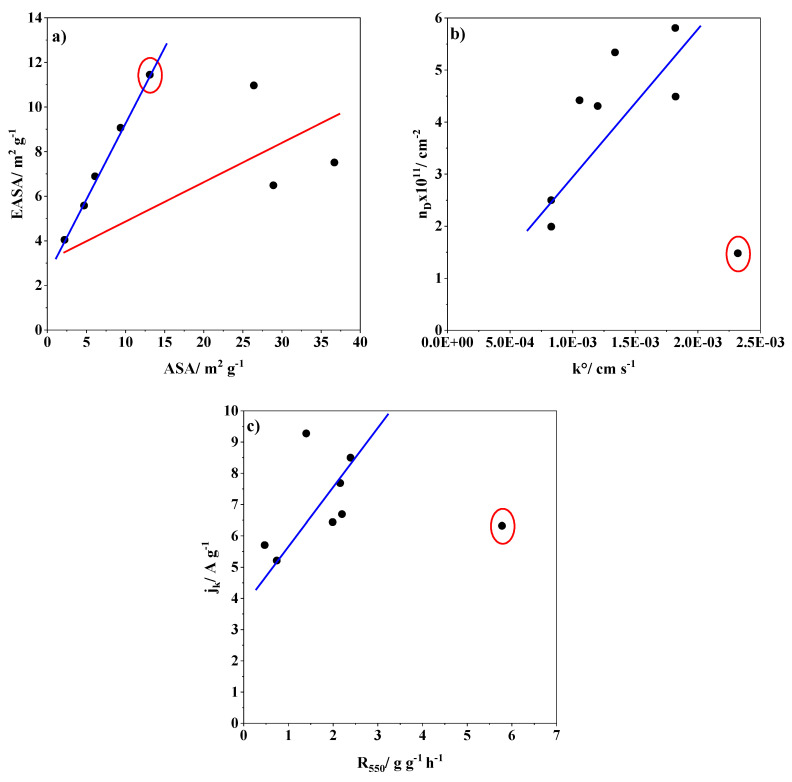
(**a**) Relationship between EASA and ASA. (**b**) Relationship between the defect density and of ESA and R_550_ of the different carbon-based materials. (**c**) The red circle surrounds the MW95_T sample.

**Table 1 nanomaterials-14-01381-t001:** Characterization of the surface and bulk composition obtained by XPS, EDX, and elemental analysis.

Samples	XPS			EDX			Elemental Analysis	
	C wt%	O wt%	N wt%	C wt%	O wt%	Other Elements wt%	C wt%	H wt%
MW99_P	98.1	1.8	0.1	97.6	1.3	1.1 *	99.3	0.0
MW99_T	98.7	1.3	0.0	100.0	0.0	0.0	99.9	0.1
MW95_P	97.8	1.4	0.8	95.6	1.6	2.8 **	89.5	0.0
MW95_T	98.8	1.2	0.0	97.6	0.9	1.5 **	96.7	0.1

* Metals detected in MW99_P: Mg, Al, Mn, and Co. ** Metals detected in MW95-based materials: Ca, Co, and Mo. Metalloid detected: Si.

**Table 2 nanomaterials-14-01381-t002:** TPD results, BET surface area, ASA, and R_550_ of the multi-wall carbon nanotubes.

Samples	CO/µmol g^−1^	CO_2/_µmol g^−1^	O/µmol g^−1^	BET Surface Area/m^2^ g^−1^	ASA/m^2^ g^−1^	R_550_/g g^−1^ h^−1^
MW99_P	1370	105	1580	195	2.35	8.1
MW99_T	389	59	507	307	9.37	1.4
MW95_P	2220	217	2654	253	38.2	4.6
MW95_T	985	117	1219	465	13.1	5.8

**Table 3 nanomaterials-14-01381-t003:** The parameters obtained from Raman spectra of the multi-walled carbon nanotubes.

Samples	Bandwidth D + D′/cm^−1^	I_D_/I_G_	I_D_/I_D′_	I_2D_/I_D_	L_sp2_/nm	L_D_/nm	n_D_ × 10^11^/cm^−2^
MW99_P	120	1.42	3.23	0.35	13.35	11.63	4.26
MW99_T	106	1.94	3.61	0.23	9.78	9.95	5.81
MW95_P	83	0.29	-	0.97	66.43	25.94	0.86
MW95_T	100	0.49	-	0.45	38.45	19.74	1.48

**Table 4 nanomaterials-14-01381-t004:** Heterogeneous electron transfer rate constant (kº) calculated through Fe(CN)_6_^3−/4−^ redox reaction, EASA obtained from Randles–Sevcik equation. The onset potential, number of electrons transferred, limiting current density, and Tafel slope are obtained for ORR reaction at carbon nanotube materials.

Sample	k^0^/cm s^−1^	EASA/ m^2^ g^−1^	$E_{onset}$/V (at −0.10 mA/cm^2^)	$E_{1/2}$/V	$n_{e^{-}}$ (at 0.7 V)	$j_{lim}$/mA cm^−2^ (at 0.4 V)	Tafel Slope/mV dec^−1^
MW99_P	1.44 × 10^−3^	4.50	0.80	0.66	2.97	−5.11	61
MW99_T	1.82 × 10^−3^	9.07	0.85	0.73	3.35	−7.52	39
MW95_P	2.11 × 10^−3^	9.07	0.82	0.70	2.80	−3.53	58
MW95_T	2.32 × 10^−3^	11.43	0.84	0.76	2.97	−3.67	47
Pt/C	-	-	0.98	0.86	3.99	−5.51	60

**Table 5 nanomaterials-14-01381-t005:** Heterogeneous electron transfer rate constant (kº) calculated through Fe(CN)_6_^3−/4−^ redox reaction, electroactive area obtained from Randles–Sevcik equation. The j_k_ value obtained from Koutecký–Levich equation for ORR, ASA, reactivity (R_550_), and defect density of the different carbon-based materials.

Sample	kº/cm s^−1^	EASA/m^2^ g^−1^	j_k_/A g^−1^	ASA/m^2^ g^−1^	R_550_/g g^−1^ h^−1^	n_D_ × 10^11^/cm^−2^
XC72	1.82 × 10^−3^	5.58 [33]	5.21	4.7 [33]	0.7 [33]	4.49
AC	1.06 × 10^−3^	7.51 [33]	7.69	36.7 [33]	2.2 [33]	4.42
YPF	1.20 × 10^−3^	10.97 [33]	6.69	26.4 [33]	2.2 [33]	4.31
CNF	1.34 × 10^−3^	6.49 [33]	6.44	28.9 [33]	2.0 [33]	5.34
Herring	8.29 × 10^−3^	4.04 [33]	5.71	2.2 [33]	0.5 [33]	2.50
SW	8.29 × 10^−3^	6.89 [33]	8.50	6.1 [33]	2.4 [33]	1.99
MW95_T	2.32 × 10^−3^	11.45	6.32	13.1	5.8	1.48
MW99_T	1.82 × 10^−3^	9.07	9.27	9.37	1.4	5.81

## Data Availability

Data are contained within the article and Appendix A.

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
