# Peer review of "Deepening the Understanding of Carbon Active Sites for ORR Using Electrochemical and Spectrochemical Techniques"

_nanomaterials, 2024, doi:10.3390/nano14171381_

Round 1

Reviewer 1 Report

Comments and Suggestions for Authors

In this work, Flores-Lasluisa et al. prepared defect-containing carbon nanotube materials by subjecting two commercial multi-walled carbon nanotubes (MWCNTs) of different purity to purification (HCl) and oxidative conditions (HNO3) and further heat treatment to remove surface oxygen groups. The as-prepared defect-containing multiwalled carbon nanotubes exhibit good catalytic performance due to the formation of carbon defects active for ORR such as edge sites and topological defects. Moreover, they exhibit good stability and methanol tolerance. Overall, this work is well prepared, thus can be published after the following issues are being addressed.

1.      In Table 1, Si is noted as a metal, which is not.

2.      XPS spectra should be carefully analyzed and the results should be compared and discussed with those from other techniques.

3.      Some recent studies should be added in the reference (e.g., 10.1021/acscatal.2c03842; 10.1038/s41467-024-50629-x).

4.      Defective carbon materials have been investigated for ORR in various studies, the authors need to highlight the new findings of this work.

Author Response

Comment 1: In Table 1, Si is noted as a metal, which is not.

Response 1: Thank you very much for the comments, we agree with you that this was a mistake and it was corrected in Table 1.

Comment 2: XPS spectra should be carefully analyzed and the results should be compared and discussed with those from other techniques.

Response 2: Thank you very much for the comment. Despite no important differences can be distinguished in the XPS spectra, we have included a paragraph discussing them, especially the O 1s spectra.

Comment 3: Some recent studies should be added in the reference (e.g., 10.1021/acscatal.2c03842; 10.1038/s41467-024-50629-x).

Response 3: Thank you very much for the comment. These two references have been included in the manuscript.

Comment 4: Defective carbon materials have been investigated for ORR in various studies, the authors need to highlight the new findings of this work.

Response 4: We have included a phrase highlighting that this work has two main objectives; the preparation of defective carbon materials from commercial carbon nanotubes electrocatalysts with an excellent ORR performance and prove that carbon defects must not be only characterized by Raman spectroscopy because this could lead to errors in understanding the catalytic behavior of carbon-based materials. Thus, it is necessary to determine EASA and ASA to support Raman spectroscopy when comparing carbon materials with different structures.

Reviewer 2 Report

Comments and Suggestions for Authors

This study focuses on the preparation of defect-containing multi-walled carbon nanotubes (MWCNTs) by subjecting commercial MWCNTs of varying purity to purification and oxidative treatments.

The physicochemical characterization of the treated carbon materials revealed significant morphological changes, including broken walls and the presence of topological defects, as observed through TEM microscopy. Raman spectroscopy analysis indicated that while carbon defects were present, not all defects contributed equally to the catalytic activity for the oxygen reduction reaction (ORR), highlighting the need for assessing active surface areas (ASA) and electrochemical active surface areas (EASA).

The study provides a detailed description of the experimental section, including the formulas used for calculations. It presents data from a wide range of methods for assessing the microstructure of the samples and their electrochemical behavior. All of this indicates the significant amount of work accomplished by the authors.

Some comments to improve the manuscript:

— It is necessary to justify why these two samples of commercial multi-walled carbon nanotubes were used for the study

— On what basis were the acid treatment conditions chosen? The authors should provide a reference to the studies on the choice of conditions or describe them in the supplementary

Author Response

Comment 1: It is necessary to justify why these two samples of commercial multi-walled carbon nanotubes were used for the study

Response 1: Thank you very much for the comment. We have included some comments in the experimental section to justify the selection of these materials.

For this purpose, it was observed the work from Gabe et al. (10.1016/j.carbon.2019.03.092) and Zhong et al. (10.1016/j.electacta.2015.12.216). Gabe et al. observed that the SWCNT showed a good ORR performance after the treatments with acids, and subsequent, thermal treatment. It was concluded that the oxidative effects can lead to structural defects. Meanwhile, Zhong et al. reported that drilling the surface of carbon nanotubes with metals produces active defects towards ORR. Then, removing the metal traces from the carbon nanotubes will create active defects in ORR. Moreover, the presence of more walls will increase the number of broken walls leading to a higher concentration of carbon defects.

Comment 2: On what basis were the acid treatment conditions chosen? The authors should provide a reference to the studies on the choice of conditions or describe them in the supplementary

Response 2: The treatment with HCl is commonly used for the purification of carbon nanotubes. Meanwhile, HNO3 is used to generate oxygen-containing functional groups and to break the walls of carbon nanotubes increasing the BET surface area. The required references have been included in the manuscript.

Reviewer 3 Report

Comments and Suggestions for Authors

The current manuscript studies on carbon active sites for  ORR using electrochemical and spectrochemical techniques. The work is interesting and systematically presented. Following revisions are suggested:

1. Schematic representation of experimental methodology can be given.

2. Any XRD characterization?

Author Response

Comment 1: Schematic representation of experimental methodology can be given.

Response 1: A schematic representation of the experimental methodology has been included in the supporting information.

Manuscript amended.

Comment 2: Any XRD characterization?

Response 2: Thank you very much for your comment. However, we did not perform XRD characterization because we already knew that the structure is graphitic (for the CNTs used in this study) and the other carbon materials like activated carbons are short range ordered materials with a graphitic-like structure. In addition, EDX detected the metal traces.

Round 2

Reviewer 3 Report

Comments and Suggestions for Authors

Suggested for accept.